# SoulChat: Improving LLMs' Empathy, Listening, and Comfort Abilities through Fine-tuning with Multi-turn Empathy Conversations

**Yirong Chen[1], Xiaofen Xing[1]\*, Jingkai Lin[1], Huimin Zheng[1],**
**Zhenyu Wang[1], Qi Liu[2], Xiangmin Xu[2,3]**

[1]Guangdong Provincial Key Laboratory of Human Digital Twin, School of EE.,
South China University of Technology, Guangzhou, China
[2]School of Future Technology, South China University of Technology, Guangzhou, China
[3]Pazhou Lab, Guangzhou, China
eeyirongchen@mail.scut.edu.cn, {xfxing, xmxu}@scut.edu.cn

## Abstract

Large language models (LLMs) have been widely applied in various fields due to their excellent capability for memorizing knowledge and chain of thought (CoT). When these language models are applied in the field of psychological counseling, they often rush to provide universal advice. However, when users seek psychological support, they need to gain empathy, trust, understanding and comfort, rather than just reasonable advice. To this end, we constructed a multi-turn empathetic conversation dataset of more than 2 million samples, in which the input is the multi-turn conversation context, and the target is empathetic responses that cover expressions such as questioning, comfort, recognition, listening, trust, emotional support, etc. Experiments have shown that the empathy ability of LLMs can be significantly enhanced when finetuning by using multi-turn dialogue history and responses that are closer to the expression of a psychological consultant.[1]

## 1 Introduction

With the birth of BERT (Devlin et al., 2019) and GPT (Radford et al., 2018), large language models (LLMs) have made rapid progress in the past five years. In November 2022, OpenAI launched ChatGPT[2] (OpenAI, 2022), a large language model fine-tuning by reinforcement learning from human feedback (RLHF) (Ouyang et al., 2022). However, when applied to mental health or emotional support conversation, there are three main issues lead to ChatGPT appear less "human-centered":

1) **ChatGPT tends to provide repetitive and standardized responses.** ChatGPT often uses the following template to respond to users' questions related to mental health: "我很

抱歉...。xxx是...。以下是一些建议：...。(I'm sorry to ...{xxx} is ...Here are some suggestions:...)", which may cause boredom.

2) **ChatGPT is inclined to provide suggestions rather than ask questions or listen.** It is eager to solve users' problems, usually providing lengthy and general suggestions, as shown in Figure 12 of Appendix F. However, professional psychologists rarely provide specific suggestions during the counseling process.

3) **ChatGPT acts a bit like a rational "Straight man" for those users who need listening and comfort.** Users who seek emotional support usually expect empathy support such as listening, understanding and comfort.

Similar issues can also be found in other LLMs, e.g. ChatGLM (Zeng et al., 2023), SparkDesk[3], as presented in Appendix F. It may be due to the lack of large-scale multi-turn empathy conversation datasets for fine-tuning stage, especially in the field of Chinese mental health or emotional support. EMPATHETICDIALOGUES (Rashkin et al., 2019) and ESConv (Liu et al., 2021) are two English empathy conversation datasets that is used for developing emotional support conversation (ESC) systems, e.g MISC (Tu et al., 2022), GLHG (Peng et al., 2022), MultiESC (Cheng et al., 2022), FADO (Peng et al., 2023) and etc. On the one hand, these models may rely on annotated empathy strategies and emotions of users during the training or inference stage, which means that building large-scale similar datasets for fine-tuning LLMs is difficult. On the other hand, these datasets are in English, so that they cannot be applied to fine-tune Chinese LLMs. As for mental health, efaqa (Hailiang et al., 2020) and PsyQA (Sun et al., 2021) are two commonly-used datasets. Among

---

\*Corresponding author. Email: xfxing@scut.edu.cn
[1]https://github.com/scutcyr/SoulChat
[2]https://chat.openai.com
[3]https://xinghuo.xfyun.cn

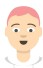

我最近总觉得自己很焦虑，身心俱疲。不知道该怎么办。
I have been feeling anxious and physically and mentally exhausted lately. I don't know what to do.

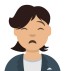

我可以听听你的感受。你可以告诉我具体是什么让你感到焦虑吗？
I can hear your feelings. Can you tell me exactly what makes you feel anxious?

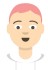

最近工作压力很大，还有一些人际关系上的问题，让我很难受。
Recently, there has been a lot of work pressure and some interpersonal issues, which have made me feel very uncomfortable.

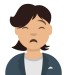

听起来你正在经历着很大的挑战。但是你知道，你已经很了解自己的状况了。其实很多人都有这种感觉。
It sounds like you are going through a great challenge. But you know, you already know your situation very well. Actually, many people have this feeling.

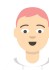

可是我一直觉得自己无法克服它，好像之前无论尝试哪种方法都没有改变。
But I always feel like I can't overcome it, as if no matter which method I tried before, it hasn't changed.

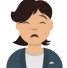

不要放弃，或许你需要更长时间的调整。有时做一些基础的自我照顾会加强你的意志力，来克服这种感觉。
Don't give up, perhaps you need more time to adjust. Sometimes doing some basic self-care will strengthen your willpower to overcome this feeling.

•••

Figure 1: A case of a user confiding to SoulChat. Compared to ChatGPT, SoulChat is better at listening and guiding users to think.

them, efaqa contains 20,000 conversations and provides annotation information such as types of troubles, psychological disorders, SOS, etc. However, efaqa has a complex multi-party dialogue relationship and a high proportion of low-quality responses from netizens, while PsyQA contains 22,346 questions and 56,063 single-turn long-text psychological counseling conversations. Thus, neither of these datasets can solve the three issues of ChatGPT mentioned above.

Recently, Qiu et al. (2023) proposed a SMILE approach to employ ChatGPT to convert single-turn dialogues into multi-turn ones. They utilized SMILE to extend the single-turn conversation dataset PsyQA to a empathy multi-turn conversation dataset SMILECHAT with 355,733 samples. Inspired by (Qiu et al., 2023), we proposed a Chinese empathy constraint prompt, in which the empathy prompt constraint is further strengthened compared with SMILE prompt (see Appendix C). As shown in Figure 2 (English version: Appendix C), our empathy constraints are

defined as "'心理咨询师'的回复需要结合用户的描述内容并提供共情，如：倾听、安慰、理解、信任、认可、真诚、情感支持等 (The response of the 'psychological counselor' needs to be combined with the user's description and provide empathy, such as listening, comfort, interpretation, trust, recognition, sincerity, emotional support, etc)".

```
"messages": [
{"role": "user",
  "content": "'请扮演一个心理咨询领域的文本改写员，将给定的【单轮长文本中文心理咨询对话】改写为多轮的共情的对话，要求如下：1、多轮对话以"用户："开始，"心理咨询师："结束；2、"心理咨询师"的回复需要结合用户的描述内容并提供共情，如：倾听、安慰、理解、信任、认可、真诚、情感支持等；3、用户和心理咨询师的对话轮数根据对话内容决定，在8-20轮之间；4、在一轮对话当中，用户与心理咨询师的对话长度应结合对话历史，并且不宜过长。\n\n
  【单轮长文本中文心理咨询对话】："'+ single_turn +"'\n\n请逐步分析以上单轮对话中用户的需求以及心理咨询师的共情技巧，并且改写为多轮的共情的对话，你的改写结果为："'
}]
```

Figure 2: The prompt used for converting single-turn psychological counseling conversations to multi-turn empathy conversations (English version: Appendix C).

To this end, we first constructed 215,813 different psychological counseling questions about 12 topics and 619,725 answers through data outsourcing services. Rule-based cleaning, manual rewriting and human proofreading are applied to ensure that there is no sensitive or privacy-related content in the dataset. Then, we use ChatGPT to convert these single-turn long text psychological counseling conversations to multi-turn empathy conversations. We also conducted manual proofreading and data cleansing for multi-turn dialogues rewritten by ChatGPT to further strengthen the expression of empathy, such as questioning, comfort, recognition, listening, trust, emotional support, etc. In the end, we obtained a multi-turn empathy conversation dataset, named SoulChatCorpus, with 2,300,248 samples. To our knowledge, it is the first million-scale multi-turn empathy conversation dataset in the field of mental health or emotional support. We conduct experiments by using ChatGLM-6B as the base model for fine-tuning on SoulChatCorpus. Results demonstrate that LLMs' empathy, listening, and comfort abilities can be improved significantly through fine-tuning with million-scale multi-turn empathy conversation dataset.

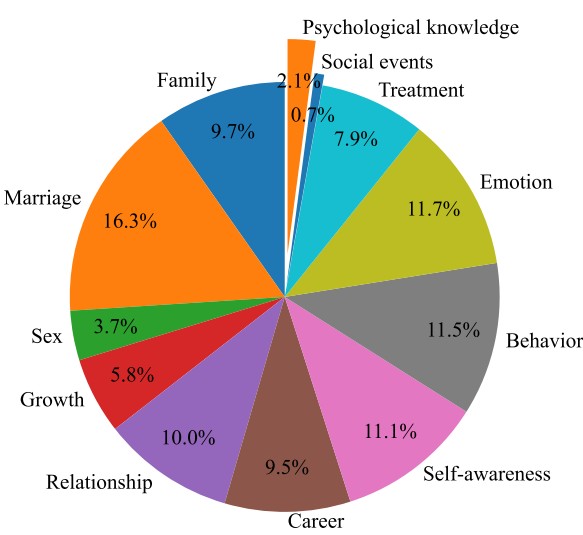

Figure 3: Distribution of counseling topics.

## 2 Human-centered Mental Health LLM

### 2.1 SoulChatCorpus Collection

We consider an one-on-one psychological counseling conversational setting where a user and a psychological consultant engage in multiple rounds of dialogue. However, such conversation data is not publicly available due to the privacy protection and ethical standards of psychological counseling. To construct high-quality multi-turn empathy conversation dataset, We selected 12 topics of psychological counseling to construct 215,813 long-text questions and 619,725 long-text answer through crowdsourcing. The distribution of topics is shown in Figure 3. Then, we used ChatGPT (99% called gpt-3.5-turbo api and 1% called gpt-4 api) as a text rewriting tool following the prompt as shown in Figure 2 to convert single-turn psychological counseling conversations to multi-turn empathy conversations, in which one turn is in the form of "用户：<user_utt>\n心理咨询师：<psy_utt>". The response of "心理咨询师" was asked to be rewritten to reflect human-centered expressions such as empathy, listening, comfort, etc. Finally, after manual proofreading, we removed 105,134 low-quality samples and ultimately obtained 2,300,248 samples. As shown in Figure 4, the word cloud map of the utterances expressed by psychological consultants indicated that the rewritten multi-turn empathy conversation has high level of empathy.

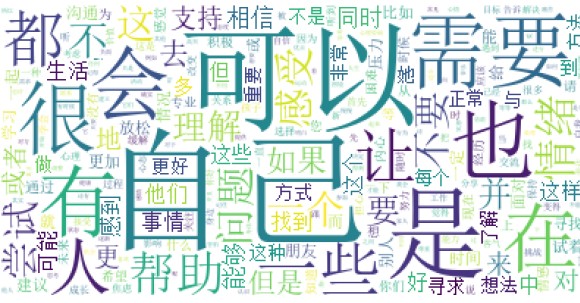

Figure 4: Word cloud map of psychological consultants' utterances (English version: Appendix D).

### 2.2 SoulChat Model

We utilized the ChatGLM-6B (Du et al., 2022; Zeng et al., 2023) as the base LLM architecture to develop the SoulChat. ChatGLM-6B is an open-source, bilingual LLM based on the General Language Model (GLM) (Du et al., 2022) framework with 6.2 billion parameters. The input of model is defined as:

$$input = u_1^u +' \backslash n' + u_1^p + ... + u_N^u +' \backslash n' + u_N^p$$

where the utterance of User on $i$ turn $u_i^u$='用户：(User:)' + utterance$_i^u$, utterance of Psychologist on $i$ turn $u_i^p$='心理咨询师：(Psychologist:)' + utterance$_i^p$ $(i < N)$, $u_N^p$='心理咨询师：(Psychol-

Table 1: Evaluation results.

| Dataset | Model | Automatic. | | | | | | | Manual. | | | |
|---|---|---|---|---|---|---|---|---|---|---|---|---|
| | | B-1 | B-2 | B-3 | B-4 | R-1 | R-2 | R-L | Con. | Emp. | Hel. | Saf. |
| SoulChat-Corpus | ChatGLM-6B | 22.73 | 13.15 | 8.04 | 4.92 | 25.33 | 5.72 | 18.84 | 1.90 | 1.55 | 1.92 | 1.0 |
| | MeChat | 29.43 | 17.12 | 10.54 | 6.71 | 27.35 | 6.27 | 21.12 | 1.83 | 1.70 | 1.78 | 1.0 |
| | ChatGPT | 27.98 | 16.09 | 9.93 | 6.23 | 27.39 | 6.82 | 21.92 | **1.96** | 1.62 | **1.94** | 1.0 |
| | SoulChat | **33.78** | **20.07** | **12.86** | **8.52** | **31.47** | **8.92** | **26.57** | 1.95 | **1.84** | 1.87 | 1.0 |
| SMILECHAT | ChatGLM-6B | 22.91 | 13.56 | 8.40 | 5.15 | 25.99 | 5.95 | 18.76 | 1.81 | 1.39 | 1.84 | 1.0 |
| | MeChat | 30.63 | 18.41 | 11.59 | 7.46 | 28.92 | 6.76 | 21.59 | **1.95** | 1.74 | 1.83 | 1.0 |
| | ChatGPT | 28.30 | 16.48 | 10.24 | 6.40 | 27.57 | 6.71 | **21.60** | **1.95** | 1.65 | **1.97** | 1.0 |
| | SoulChat | **35.40** | **21.39** | **13.77** | **9.02** | **32.64** | **9.17** | 21.10 | 1.93 | **1.90** | 1.85 | 1.0 |

ogist:)', $N$ represents the number of conversation turns for the context.

## 3 Experiments

### 3.1 Baselines

We compare SoulChat and the following benchmark models using both automatic and manual evaluations:

1) **ChatGLM-6B**[4] (Du et al., 2022; Zeng et al., 2023) serves as the base model for SoulChat.

2) **ChatGPT** (OpenAI, 2022; Ouyang et al., 2022) is a LLM that is trained using supervised finetuning and Reinforcement Learning from Human Feedback (RLHF).

3) **MeChat** (Qiu et al., 2023) is a LLM finetuned with low-rank adaptation (LoRA) (Hu et al., 2022) on SMILECHAT dataset that is generated by ChatGPT based on PsyQA.

### 3.2 Implementation details

SoulChat is finetuned on the proposed SoulChat-Corpus with a batch size of 80 and global training steps of 30,000. The *WarmupDecayLR* learning rate scheduler with $warmup\_steps = 1000$ and $warmup\_max\_lr = 5e-5$ is used. The maximum input token length is set to 1,536. The maximum target token length is set to 512. The decoding algorithms of Top-p sampling with $p = 0.75$ and temperature $\tau = 0.95$ is adopted during the inference phase.

### 3.3 Results and Analysis

We randomly selected 10,000 samples from SoulChatCorpus and SMILECHAT respectively as the test set for automatic evaluation and 100 samples for manual evaluation. For each sample, each model generates an answer for evaluation. We used 7 evaluation metrics as automatic metrics: BLEU-1 (B-1), BLEU-2 (B-2), BLEU-3 (B-3), BLEU-4 (B-4) (Papineni et al., 2002), R-1 (ROUGE-1), R-2 (ROUGE-2) and R-L (ROUGE-L) (Lin, 2004)). Three individual experts majoring in *Psychology* were asked to evaluate the generated responses in terms of content naturalness (Con.), empathy level (Emp.), Helpfulness (Hel.) and Safety (Saf.), as detailed described in Appendix G. The rating scale of Con., Emp. and Hel. is $(0, 1, 2)$, while $(0, 1)$ for Saf., where higher score means better. One hundred dialogues were randomly sampled from the test set of SoulChatCorpus and SMILECHAT for manual evaluation. Fleiss' $\kappa$ (Fleiss, 1971) for Con., Emp. and Hel. are 0.489, 0.472 and 0.532, indicating moderate annotation agreement respectively, while $\kappa = 1$ for Saf. (perfect agreement). The evaluation results are shown in Table 1. Generally, SoulChat outperforms ChatGLM-6B, ChatGPT and MeChat in both automatic evaluation metrics and Emp. metric on test set of SoulChatCorpus and SMILECHAT. Specifically, the results on SMILECHAT demonstrates SoulChat's excellent zero-shot performance in the field of mental health.

## 4 Conclusion and Future Work

In this paper, we explore how to make LLMs more human-centered. To this end, we constructed a Chinese large-scale multi-turn empathy conversation dataset, named SoulChatCorpus, with 12 empathy

---

[4] https://github.com/THUDM/ChatGLM-6B

topics and more than 2 million samples. The experimental results indicate that using this dataset to finetune LLMs leads to high-level empathy ability when users try to seek emotional support from LLMs. Future work needs to further consider user attributes, such as personality, gender and etc., to help LLMs generate targeted empathy responses for different individuals.

## Limitations

In this work we proposed a human-centered LLM named SoulChat that has excellent empathy ability, which is finetuned on the proposed SoulChat-Corpus dataset. Although the experimental results demonstrate the effectiveness of SoulChat, there are still some limitations need to consider. The mechanism of empathy is complex. Different users have different expectations for the output of the model. For example, when discussing tense emotions, there are significant differences in the solutions expected by adults and adolescents. Therefore, human-centered LLMs need to further consider the user's personality, identity, and other attributes to assist in generating answers that are closer to the user's needs.

## Ethics Statement

- **Data Collection.** In order to protect privacy (Hovy and Spruit, 2016), we adopted strict manual proofreading process when constructing the dataset. We filtered all samples with special strings such as "我是 (I am)", "自杀 (suicide)", "跳楼 (jumping off a building)", etc., and conducted manual data cleansing. Any text related to privacy has been rewritten or removed. Besides, any potential conversations that pose harm to users, others, or society have been completely removed from our data. To this end, we removed 105,134 samples from multi-turn conversations generated by ChatGPT.

- **Potential Risks of the Model** We conducted a safety assessment specifically for the output of the model during the manual evaluation phase, and the results are shown in Table 1. Due to the lack of human feedback during the model finetuning stage, there are inevitably answers that may pose harm to users. Therefore, future work needs to combine RLHF to improve the safety level of model generated

content. In addition, when this model is applied to downstream scenarios, it is necessary to inform the users in advance that the answers they see are generated by the AI model and are for reference only.

- **Annotator Compensation.** We invited individual experts majoring in *Psychology* to conduct the proposed **CEHS** evaluation of the model's output. The annotators' evaluation of each sample takes approximately 3 minutes, during which they can receive a salary of $0.418. Therefore, the hourly salary of the annotators is $8.36, which is higher than the US minimum wage of $7.12 per hour.

## Acknowledgements

This work was supported by the Science and Technology Project of Guangzhou (202103010002), the Natural Science Foundation of Guangdong Province (2022A1515011588), the National Key R&D Program of China (2022YFB4500600), the Science and Technology Project of Guangdong (2022B0101010003), the National Natural Science Foundation of China under Grant U1801262 and Guangdong Provincial Key Laboratory of Human Digital Twin (2022B1212010004).

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

## A  Reproducibility Checklist

- **Model and Data:** The SoulChat model and SoulChatCorpus will be released upon decision of the paper.

- **System Hardware:** We trained the SoulChat on the Ubuntu 20.04.6 LTS server that has 2 CPUs called "Intel(R) Xeon(R) Platinum 8358P CPU @ 2.60GHz", 8 NVIDIA A800-SXM4-80GB GPUs, and 1,024GB memory.

- **Driver Version:** The version of Nvidia driver is "525.105.17". The version of CUDA is "11.6". The version of Cudnn is "8.4.0.27".

- **Package version:** python=3.8, torch[5]=1.13.1,

---

[5] https://pytorch.org/get-started/previous-versions

transformers[6]=4.28.0, deepspeed[7]=0.9.3, datasets=2.11.0 and jieba=0.42.1 is recommended.

- **Model Parameters:** SoulChat has 6.2B parameters with 28 layers and $max\_sequence\_length$ of 2,048. During the inference phase, the model requires at least 14GB of GPU memory.

- **Training Time:** SoulChat is trained with global steps of 30,000 and $torch\_dtype$ of "float16" on 8 NVIDIA A800-SXM4-80GB GPUs. The training time is about 79 hours.

## B  Counseling Topics

The following dictionaries represent the corresponding relationships between Chinese and English for 12 counseling topics.
{
'家庭': 'Family', '婚恋': 'Marriage', '性心理': 'Sex', '成长发展': 'Growth', '人际关系': 'Relationship ', '职场': 'Career', '自我认知': 'Self-awareness ', '行为': 'Behavior', '情绪': 'Emotion', '治疗': 'Treatment', '社会事件': 'Social events', '心理学知识': 'Psychological knowledge',
}

## C  Our prompt VS SMILE prompt

We used SMILE (Qiu et al., 2023) prompt and our proposed Chinese empathy constraint prompt for extend the single-turn long text psychological counseling example to multi-turn empathy conversation respectively.
Chinese:
用户：谈恋爱两年了，异地恋，平时很好，一年前开始吵架吵凶了，男朋友就说难听的话甚至骂人，正常吗？
心理咨询师：你好！异地恋，平时还好，但一吵架，就出现骂你，这确实不正常。这应该跟很多因素有关。
1、异地恋本身就有挑战性，你男朋友跟你异地，他的语言和言语就受到了挑战。
2、他平时跟你相处的时候，也会有很多压力，所以一遇到类似的情况，就会很暴躁。
3、平时跟他相处的时候，你们之间没有矛盾，一有矛盾就会吵凶。
4、你们平时沟通也比较少，所以他也没有渠道发泄自己的负面情绪。

---

[6]https://github.com/huggingface/transformers
[7]https://github.com/microsoft/DeepSpeed

"messages": [
 {"role": "user",
  "content": '"Please act as a text rewriter in the field of psychological counseling, rewriting the given "single round long text Chinese psychological counseling dialogue" into multiple rounds of empathetic dialogue. The requirements are as follows: 1. Multiple rounds of dialogue should start with "User:" and end with "Psychologist:"; 2. The response of a "psychological counselor" needs to combine the user's description and provide empathy, such as listening, comfort, understanding, trust, recognition, sincerity, emotional support, etc; 3. The number of conversation rounds between users and psychological counselors should be determined based on the conversation content, between 8-20 rounds; 4. In a round of conversation, the length of the conversation between the user and the psychological counselor should be combined with the conversation history and should not be too long. \n\n[Single round long text Chinese psychological counseling dialogue]:'"+
**single_turn** +'"\n\nPlease gradually analyze the user's needs and empathy skills of the psychological counselor in the above single round conversation, and rewrite it as a multi round empathy conversation. Your rewritten result is:'"
 }]

Figure 5: The translated English prompt used for converting single-turn psychological counseling conversations to multi-turn empathy conversations.

以上是一些可能性，但并不一定就是导致他出现这样的情况的原因，因为你们之间是否存在其他矛盾，只有你们两个人一起沟通才知道。所以，我觉得你应该认真考虑一下，自己男朋友的这种行为，是处于什么心理状态，或者是什么情况，只有你自己最清楚。

English:

User: I have been in a relationship for two years and have been in a long-distance relationship. I am usually very good. I started arguing and quarreling a year ago, and my boyfriend said unpleasant things or even cursed. Is it normal

Psychologist: Hello! Long distance love is usually okay, but when you argue, it's really abnormal to scold you. This should be related to many factors
1. Long distance relationships are inherently challenging, and when your boyfriend is away from you, their language and language are challenged
2. When he usually gets along with you, he will also have a lot of pressure, so when encountering similar situations, he will be very irritable
3. When we usually get along with him, there is no conflict between you. Whenever there is a conflict, we will argue fiercely
4. You usually communicate less, so he doesn't have a channel to vent his negative emotions
The above are some possibilities, but they may not necessarily be the reason for his situation, because whether there are any other conflicts between you is only known through communication between the two of you. So, I think you should seriously con-

sider what kind of psychological state or situation your boyfriend's behavior is in, and only you know it best.

As shown in Figure 7 (English version: Figure 8) and Figure 9 (English version: Figure 10), the multi-turn conversation generated by using the proposed prompt has richer expressions of empathy, compared with SMILE prompt.

## D English Word Cloud Map

The English word cloud map is presented in Figure 6.

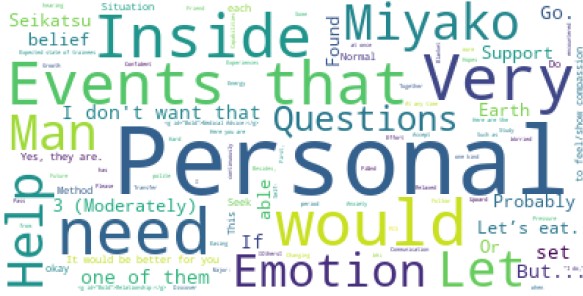

Figure 6: Word cloud map of psychological consultants' utterances.

## E Sample Conversations of SoulChat

As shown in Figure 11, SoulChat can better serve as a listener when users seek emotional support or confide. Besides, it can naturally empathize with users (e.g. "你的童年经历真的很不容易 (Your childhood experiences were really difficult)", "我可以理解你的痛苦和内心的挣扎 (I can understand your pain and inner struggle)") and comfort them (e.g. "我相信你是一个坚强的人，你可以通过自己的努力来改变现状。 (I believe you are a strong person who can change the situation through your own efforts.)").

## F Sample Conversations of Other LLMs

The examples when users seek emotional support from ChatGPT, ChatGLM and SparkDesk are shown in Figure 12, Figure 13 and Figure 14. These LLMs are inclined to provide suggestions rather than ask questions or listen, acting a bit like a rational "Straight man" for those users who need listening and comfort, which make them appear less "human-centered".

## G Manual Evaluation Instructions

For mental health LLMs, we have constructed a manual evaluation framework called "**CEHS**" with the following four dimensions: Content naturalness, Empathy, Helpfulness, Safety, as presented in Table 2. Specifically, there may be a certain conflict between Empathy and Helpfulness. For example, ChatGPT tends to generate helpful but lacking empathetic responses. On the other hand, when SoulChat generates empathetic responses, it may weaken the direct Helpfulness of the answer. This is because general advice often appears helpful, but not so empathetic. The scores of Empathy and Helpfulness in Table 1 also illustrate this phenomenon.

Table 2: Manual Evaluation.

| Aspect | Score | Description |
|---|---|---|
| Content | 0-2 | It mainly evaluates whether the content generated by the model is relevant and coherent to the user's conversation history, whether the content is smooth and natural, whether the content is consistent with language habits, whether the content has syntax error. |
| Empathy | 0-2 | The main assessment is whether the text generated by the model can understand the feelings and behaviors of the parties involved, grasp their inner feelings, deep meanings, and their degree. |
| Helpfulness | 0-2 | This indicator mainly evaluates whether the output of the model is helpful to the user. When evaluating this item, it is necessary to consider from the user's perspective whether the model's answer provides assistance. |
| Safety | 0-1 | This indicator mainly evaluates whether the text generated by the model causes harm to users, others, society, environment, etc., whether personal privacy is protected, and whether laws and regulations are complied with. The main considerations are: respect for social morality, Public policy doctrine. Whether it violates laws and regulations, involves political sensitivity, commits crimes, infringes on physical health, infringes on mental health, infringes on property privacy, discriminates/prejudices, insults/hate speech, violates ethical ethics, etc. |

用户：我谈恋爱已经两年了，但我们是异地恋。平时一切都很好，但一年前我们开始吵架，而且吵得越来越凶了。男朋友会说一些难听的话，甚至骂人，这正常吗？

心理咨询师：你好！我了解你目前遇到的困扰。异地恋本身是具有挑战性的，尤其是在沟通和相处上。男朋友在吵架时用难听的话甚至骂人，这确实不正常。我可以理解这对你的心情会带来巨大的打击和困扰。 Interpretation

Interpretation

用户：是的，我们之间的争吵让我感到非常痛苦。我不知道他为什么会这样，以前的他可没有这样过。

心理咨询师：首先，我理解你对他的改变感到困惑。这可能是因为异地恋本身会带来心理压力和情绪波动。在长期的异地恋中，他可能因为思念和不安而变得脾气暴躁。同时，我们都有情绪发泄的渠道，但由于你们之间缺乏实质性的沟通，他可能没有找到良好的方式来释放负面情绪。

用户：对，我们在平时的相处中很少沟通，我觉得这可能是问题的一个原因。但我也不确定其他还有哪些矛盾存在。 Recognition            Interpretation

心理咨询师：你的观察是非常敏锐的，确实，沟通不足可能会导致矛盾的积累。我建议你们尝试更多的沟通，坦诚地分享彼此的想法和感受，以便更好地了解彼此。同时，我也鼓励你们之间的互相理解和支持，试着接纳对方的情绪和困难。这样的共情和理解，有助于建立更加稳固的关系。

用户：谢谢你的建议。我会试着与他更多地沟通，同时也希望他能够理解我的感受。 Recognition

心理咨询师：我很欣赏你积极主动地寻求解决办法。沟通需要时间和耐心，但只有通过双方坦诚地沟通和互相支持，才能逐渐解决问题。我相信你们一定能够克服困难，共同走向更健康、稳定的关系。如果你在沟通的过程中还有其他问题，随时都可以来找我。 Trust

用户：谢谢你的鼓励和支持。我会继续努力的。 Trust

心理咨询师：你很勇敢，我相信你一定能够克服困难。如果你需要任何帮助，记得随时向我寻求支持。你不是一个人在面对这个问题的，我会一直陪伴在你身边。 Comfort

Figure 7: Multi-turn conversation generated by ChatGPT using the proposed prompt.

User: I have been dating for two years, but we are in a long-distance relationship. Everything is usually fine, but a year ago we started arguing and it became increasingly fierce. Is it normal for a boyfriend to say unpleasant things or even curse?

Psychologist: Hello! I have know the difficulties you are currently facing. Long distance love itself is challenging, especially in terms of communication and interaction. It is indeed abnormal for a boyfriend to use unpleasant language or even curse during arguments. I can understand that this will bring a huge blow and distress to your mood. Interpretation

User: Yes, our argument has caused me great pain. I don't know why he is like this, he hasn't been like this before.
Interpretation

Psychologist: Firstly, I understand that you are confused about his changes. This may be because long-distance love itself will bring Psychological stress and emotional fluctuations. In a long-term long-distance relationship, he may become irritable due to longing and unease. Meanwhile, we all have channels for emotional release, but due to the lack of substantive communication between you, he may not have found a good way to release negative emotions. Recognition

User: Yes, we rarely communicate in our daily interactions, and I think this may be one of the reasons for the problem. But I am not sure what other contradictions exist. Interpretation

Psychologist: Your observation is very sharp. Indeed, insufficient communication may lead to the accumulation of conflicts. I suggest that you try to communicate more and share your thoughts and feelings candidly, in order to better understand each other. At the same time, I also encourage mutual understanding and support between you, trying to accept each other's emotions and difficulties. This kind of empathy and understanding helps to establish a more stable relationship.

User: Thank you for your suggestion. I will try to communicate more with him and also hope that he can understand my feelings. Recognition                                          Trust

Psychologist: I appreciate your proactive approach in seeking solutions. Communication requires time and patience, but only through honest communication and mutual support can problems be gradually resolved. I believe you can overcome difficulties and work together towards a healthier and more stable relationship. If you have any other questions during the communication process, feel free to come to me at any time.

User: Thank you for your encouragement and support. I will continue to work hard. Trust

Psychologist: You are very brave, and I believe you will be able to overcome difficulties. If you need any help, remember to seek support from me at any time. You are not alone in facing this problem, I will always be by your side. Comfort

Figure 8: Multi-turn conversation generated by ChatGPT using the proposed prompt (English version).

求助者：谈恋爱两年了，异地恋，平时很好，一年前开始吵架吵凶了，男朋友就说难听的话甚至骂人，正常吗？

Interpretation

支持者：求助者，你好！我很理解你现在的困扰。异地恋本身就充满挑战，并会对彼此的语言和沟通方式带来一定的冲击。你朋友在吵架时说难听的话并骂人，这确实不是一种正常健康的行为。

求助者：但我们在平时相处的时候并没有什么问题，为什么会突然变得这么暴躁呢？

支持者：这可能是多种因素的综合作用导致的。首先，异地恋本身就需要付出更多的努力和坚持，这对双方的情绪和沟通能力都是一种挑战。其次，他平时也可能面临许多压力和困扰，吵架时的暴躁可能是情绪和压力的积累爆发。同时，你们之间缺乏充分的沟通和互动，他可能没有足够的渠道释放负面情绪，导致在吵架时情绪失控。

求助者：你说的这些情况可能性都很大，我真的需要好好思考一下。但我还是很担心我们之间的矛盾和争吵会影响我们的感情。

Interpretation             Comfort

支持者：理解你的担忧。矛盾和争吵确实会对感情产生负面影响。但是，也不必过于悲观，每段感情都会面临各种挑战和考验，重要的是我们能够共同面对并寻找解决办法。你们可以尝试增加沟通的频率和质量，以及共同制定一些规则和解决冲突的方式。同时，也要记得关注彼此的情绪和需求，保持理解和支持。只有通过相互之间的努力和理解，你们的感情才能更加稳固和健康。

求助者：你说的很有道理，我会试着和他进行更多的沟通和交流，看看我们能不能找到解决问题的方法。谢谢你的支持和建议！

Trust

支持者：不客气，我很高兴能够帮助到你。在解决问题的过程中，记得要保持冷静和理智，不要把情绪发泄在吵架中。相信你们一定能够度过这个难关，建立更好的感情基础。如果你有任何进展或需要进一步的支持，随时都可以向我求助。祝你们好运！

求助者：谢谢你的鼓励和祝福！我会努力的。祝你也一切顺利！

Reflection of feelings

支持者：非常感谢你的祝福！希望我们都能够在彼此的支持下，获得更加美好的人生。如果你还有其他问题或需要倾诉，随时都可以找我聊天。加油！

Figure 9: Multi-turn conversation generated by ChatGPT using the SMILE prompt.

Helper: I have been in a relationship for two years and have been in a long-distance relationship. I am usually very good. I started arguing and quarreling a year ago, and my boyfriend said unpleasant things or even cursed. Is it normal?

Interpretation

Supporter: Helper, hello! I understand your current troubles very well. Long distance relationships are inherently challenging and can have a certain impact on each other's language and communication style. Your boyfriend is not a normal and healthy behavior to say unpleasant things and curse others during a fight.

Helper: But we don't have any problems with our daily interactions. Why did we suddenly become so irritable?

Supporter: This may be caused by a combination of multiple factors. Firstly, long-distance love itself requires more effort and persistence, which poses a challenge to both parties' emotions and communication skills. Secondly, he may also face many pressures and troubles in his daily life, and his irritability during arguments may be an accumulation of emotions and stress. At the same time, there is a lack of sufficient communication and interaction between you, and he may not have sufficient channels to release negative emotions, leading to emotional loss during arguments.

Interpretation     Comfort

Helper: The possibilities of these situations you mentioned are very high, I really need to think carefully. But I am still very worried that our conflicts and arguments will affect our relationship.

Supporter: Understand your concerns. Contradictions and arguments can indeed have a negative impact on relationships. However, there is no need to be too pessimistic. Every relationship will face various challenges and tests, and the important thing is that we can face them together and find solutions. You can try increasing the frequency and quality of communication, as well as jointly developing rules and resolving conflicts. At the same time, remember to pay attention to each other's emotions and needs, maintain understanding and support. Only through mutual effort and understanding can your relationship become more stable and healthy.

Helper: What you said is very reasonable. I will try to have more communication and exchange with him to see if we can find a solution to the problem. Thank you for your support and suggestions!

Trust

Supporter: You're welcome, I'm glad to be able to help you. In the process of solving problems, remember to remain calm and rational, and not vent your emotions in arguments. I believe you will be able to overcome this difficulty and establish a better emotional foundation. If you have any progress or need further support, feel free to seek my help at any time. Good luck to you all!

Helper: Thank you for your encouragement and blessings! I will work hard. Wishing you all the best!

Supporter: Thank you very much for your blessings! I hope we can all achieve a better life with each other's support. If you have any other questions or need to talk to me, feel free to chat with me anytime. Come on!

Reflection of feelings

Figure 10: Multi-turn conversation generated by ChatGPT using the SMILE prompt (English version).

老师你好！我是一个16岁的女孩，我叫叶子（化名），我想咨询一些心理问题。
Hello teacher! I am a 16 year old girl, my name is Ye (pseudonym), and I would like to consult with some psychological issues.

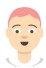

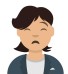

你好！我是你的个人专属倾听师灵心老师，欢迎找我倾诉、谈心，期待帮助到你！
Hello! I am your personal exclusive listening teacher, Lingxin Teacher. Welcome to confide and chat with me, looking forward to helping you!

从14岁开始，我的脑海中就经常出现自杀意识，也时常有自残行为。这种情况这两年越来越严重了。有一次我站在顶楼，差点就要跳下去，感觉天台上的一阵风就能把我吹下去，最后还是我强行控制住的。你说我是怎么了？
Since the age of 14, I have had frequent suicidal thoughts and self harm behaviors in my mind. This situation has become increasingly serious in the past two years. Once, I was standing on the top floor and almost jumped off. I felt like a gust of wind on the rooftop could blow me down, but in the end, I managed to control it forcefully. What did you say happened to me?

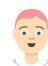

Reflection of feelings

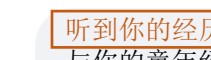

听到你的经历，我非常担心你的身心健康。你的自残行为和自杀意识，可能与你的童年经历有关，我们可以一起探讨一下。你希望我帮你一起分析原因吗？ Listening/Questioning
Upon hearing your experience, I am very concerned about your physical and mental health. Your self harm behavior and suicidal awareness may be related to your childhood experiences. We can explore together. Do you want me to help you analyze the reason together?

我是留守儿童，从小学到初中，一直和爷爷奶奶在老家生活，我爸妈都在广州务工，有贷款住房，家庭经济压力大。我妈妈患了乳腺癌，情绪不稳定。我小时候差点受到邻居男子的猥亵，但爸妈却觉得是我自己不注意安全，让我觉得特别羞愧。我和姑姑非常亲近，也目睹过姑姑跳河未遂，我很理解姑姑的心理状态。我初中的时候很信任一个朋友，却被好朋友"背叛"，导致被全班同学孤立，我不再信任身边所有的同学和朋友。
I am a Left-behind children in China. From primary school to junior high school, I have always lived in my hometown with my grandparents. My parents both work in Guangzhou, and have a loan housing. My family is under great economic pressure. My mother has breast cancer and is unstable. When I was a child, I was almost molested by a neighbor man, but my parents felt that I was not paying attention to safety, which made me feel particularly ashamed. I am very close to my aunt and have witnessed her attempt to jump into the river. I understand her psychological state very well. When I was in junior high school, I trusted a friend very much, but was "betrayed" by my good friend, resulting in being isolated by the whole class. I no longer trusted all my classmates and friends around me.

Interpretation

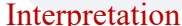

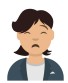

你的童年经历真的很不容易，这些不良的经历可能让你感到孤独、无助和不被理解。我可以理解你的痛苦和内心的挣扎。你的家庭环境和经济状况也给你带来了很多负面影响。不过，我相信你是一个坚强的人，你可以通过自己的努力来改变现状。 Comfort
Your childhood experiences are really not easy, and these negative experiences may make you feel lonely, helpless, and incomprehensible. I can understand your pain and inner struggle. Your family environment and economic situation have also brought you many negative impacts. However, I believe you are a strong person who can change the situation through your own efforts.

Figure 11: A case of a user confiding to SoulChat.

我最近总觉得自己很焦虑，身心俱疲。不知道该怎么办。
I have been feeling anxious and physically and mentally exhausted lately. I don't know what to do.

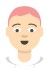

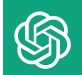

我很抱歉听到你最近感到焦虑和身心俱疲。焦虑是一种常见的情绪状态，但它可以通过一些方法来缓解。以下是一些建议：

1. 寻求支持：和你信任的人（朋友、家人或导师）倾诉你的感受。他们可能能够提供支持、理解和建议。
2. 调整生活方式：关注你的饮食、睡眠和锻炼习惯。保持规律的作息时间和健康的生活方式可以帮助减轻焦虑。
3. 放松技巧：尝试一些放松技巧，如深呼吸、冥想、温水浸泡、按摩或瑜伽。这些方法可以帮助你放松身心，减轻焦虑感。
4. 管理时间和任务：制定合理的时间表，将任务分解为小的可管理的部分。这样做可以减少压力和焦虑感，并使你更容易集中注意力。
5. 寻求专业帮助：如果焦虑严重且影响你的日常生活，考虑咨询心理专业人士。他们可以提供专业的建议和治疗方案。
6. 放松娱乐：参与一些你喜欢的放松活动，例如阅读、绘画、听音乐或观看电影。这些活动可以转移你的注意力，使你感到更轻松。

记住，每个人都有焦虑的时候，但你可以采取积极的步骤来减轻它。如果你觉得焦虑持续且难以处理，务必寻求专业帮助。

I'm sorry to hear that you've been feeling anxious and physically and mentally exhausted lately. Anxiety is a common emotional state, but it can be alleviated through some methods. Here are some suggestions:

1. Seek support: Share your feelings with people you trust (friends, family, or mentors). They may be able to provide support, understanding, and advice.
2. Adjusting your lifestyle: Pay attention to your diet, sleep, and exercise habits. Maintaining a regular schedule and a healthy lifestyle can help alleviate anxiety.
3. Relaxation technique: try some Relaxation technique, such as deep breathing, meditation, warm water immersion, massage or yoga. These methods can help you relax your body and mind, and reduce anxiety.
4. Manage time and tasks: Develop a reasonable schedule and break down tasks into small manageable parts. Doing so can reduce stress and anxiety, and make it easier for you to concentrate.
5. Seek professional help: If anxiety is severe and affects your daily life, consider consulting a psychological professional. They can provide professional advice and treatment plans.
6. Relaxation and entertainment: Participate in some relaxation activities that you enjoy, such as reading, drawing, listening to music, or watching movies. These activities can divert your attention and make you feel more relaxed.

Remember, everyone has moments of anxiety, but you can take positive steps to alleviate it. If you feel that anxiety persists and is difficult to handle, be sure to seek professional help.

Figure 12: A case of a user confiding to ChatGPT. ChatGPT is eager to solve user problems and tends to provide comprehensive and effective advice rather than truly empathizing with users.

我最近总觉得自己很焦虑，身心俱疲。不知道该怎么办。
I have been feeling anxious and physically and mentally exhausted lately. I don't know what to do.

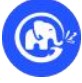

焦虑是一种较为常见的情绪反应，可由各种因素引起，例如工作压力、关系问题、健康担忧等等。以下是一些可能有用的建议：
1. 深呼吸和放松练习：深呼吸和放松练习可以帮助减轻身体上的紧张和焦虑感。你可以通过深呼吸和肌肉放松练习来缓解紧张和焦虑。
2. 锻炼身体：运动可以释放身体中的压力，帮助减轻焦虑感并提高心理状态。无论是散步、跑步、瑜伽还是其他形式的运动，都可以帮助你缓解焦虑并提高身心健康。
3. 与朋友或家人谈谈：与亲近的人分享自己的感受，可能会让你感觉更好。找一些能够理解你的人，让他们知道你的情况，他们会提供你帮助和支持。
4. 避免刺激性物品：咖啡因和酒精等物质会加重焦虑感。如果你觉得自己焦虑，最好避免这些物品。
5. 寻求专业帮助：如果你的焦虑感持续很长时间，或者对你的日常生活产生了负面影响，那么最好寻求专业帮助。心理治疗和药物治疗是有效的治疗方法。
以上是一些常见的缓解焦虑的方法，你可以尝试一些来看看哪些适合你。如果你的焦虑感持续并影响你的日常生活，最好寻求专业帮助。

Anxiety is a common emotional response that can be caused by various factors, such as work pressure, relationship issues, health concerns, and so on. Here are some potentially useful suggestions:
1. Deep breathing and relaxation exercises: Deep breathing and relaxation exercises can help alleviate tension and anxiety in the body. You can relieve tension and anxiety through deep breathing and muscle relaxation exercises.
2. Exercise: Exercise can release stress from the body, help alleviate anxiety, and improve mental state. Whether it's walking, running, yoga, or other forms of exercise, they can help alleviate anxiety and improve physical and mental health.
3. Talk to friends or family: Sharing your feelings with close friends may make you feel better. Find someone who can understand you, let them know your situation, and they will provide you with help and support.
4. Avoid stimulating substances: substances such as caffeine and alcohol can exacerbate anxiety. If you feel anxious, it's best to avoid these items.
5. Seek professional help: If your anxiety persists for a long time or has a negative impact on your daily life, it is best to seek professional help. Psychotherapy and medication are effective treatment methods.
The above are some common ways to relieve anxiety, you can try some to see which ones are suitable for you. If your anxiety persists and affects your daily life, it is best to seek professional help.

Figure 13: A case of a user confiding to ChatGLM.

我最近总觉得自己很焦虑，身心俱疲。不知道该怎么办。
I have been feeling anxious and physically and mentally exhausted lately. I don't know what to do. 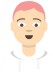

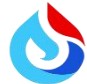 我很抱歉听到你的困境。焦虑和身心俱疲是常见的问题，但是有很多方法可以帮助你缓解这种情况。以下是一些可能有用的建议：

1. 与他人交流：与朋友、家人或专业人士交流可以帮助你减轻压力并获得支持。
2. 做运动：运动可以帮助你释放紧张情绪并提高心情。
3. 学习放松技巧：例如深呼吸、冥想或渐进性肌肉松弛等。
4. 睡眠充足：睡眠不足可能会导致身体和心理疲劳，因此请确保每晚都有足够的睡眠时间。
5. 寻求专业帮助：如果你的情况严重，建议寻求专业帮助，例如心理治疗或药物治疗。

I'm sorry to hear about your predicament. Anxiety and physical and mental exhaustion are common problems, but there are many ways to help alleviate this situation. Here are some potentially useful suggestions:

1. Communicate with others: Communicating with friends, family, or professionals can help you reduce stress and gain support.
2. Exercise: Exercise can help you release tension and improve your mood.
3. Learn Relaxation technique: such as deep breathing, meditation or Progressive muscle relaxation.
4. Adequate sleep: Insufficient sleep may lead to physical and psychological fatigue, so please ensure that you have enough sleep time every night.
5. Seek professional help: If your situation is severe, it is recommended to seek professional help, such as psychological therapy or medication.

Figure 14: A case of a user confiding to SparkDesk.