# OpenReview forum: "SoulChat: Improving LLMs' Empathy, Listening, and Comfort Abilities through Fine-tuning with Multi-turn Empathy Conversations"
_EMNLP/2023/Conference — EMNLP 2023 Findings_

### Official Review · Reviewer_1XLJ · 2023-07-26

**Soundness:** 3

**Excitement:**

4: Strong: This paper deepens the understanding of some phenomenon or lowers the barriers to an existing research direction.

**Paper Topic And Main Contributions:**

The paper describes two main contributions. On the one hand, it introduces SoulChatCorpus as a new data resource. The corpus contains 2.3 million samples of utterances in multi-turn empathy conversation. To my understanding, the corpus contains (constructed) dialogs in 12 topics/domains of psychological counseling. The corpus language is Chinese.

On the other hand, the authors describe SoulChat, a conversational system which is based on ChatGLM-6B that was fine-tuned using the SoulChatCorpus. SoulChat is used to demonstrate in an NLP engineering experiment the advantage for empathy conversations when a modle is fine-tuned with a large amount of data.

A minor, but still helpful, contribution is the overview on empathy-related conversation datasets.

**Questions For The Authors:**

Question A: fig. 10 shows the translation of Fig. 4. and the word cloud should show that the SoulChat conversations have a high level of empathy. Can you elaborate in which way the words in the word cloud a represent a high level of empathy?
Question B: How many dialogs have been generated with each model and for each dataset in the evaluation?
Question C: Can you provide Krippenbach's alpha to check the inter-annotator agreement?
Question D: Why have the three experts rated examples out of the dataset test sets and not examples generated by the models? (cp. 201-203)
Question E: The experts are students of Psychology? In which stage of their studies they have been at the time of the evaluation?
Question G: Can you provide the meanings of u, n, p in the input definition (above line 160) and translate everything in the lines 160+161 and 148-150?
Question H: The input lengths in the lines 185 and 186 are measured in tokens, words, or signs?
Question I: Can you explain how the evaluation was done? (what was generated by the models on the basis of which input, how have you applied the automatic measures - especially in case of multi-turn dialogs).

**Reasons To Accept:**

- The paper well motivates why the construction of the SoulChatCorpus was necessary by giving an interesting overview on the existing datasets in the domain of emphatic datasets
- The idea to use ChatGPT (and probably LLMs in general) to create multi-turn-dialogs out of single-turn conversations is smart
- An evaluation study with three baseline models (pure ChatGLM-6B, MeChat, and ChatGPT) and as baseline dataset (SMILECHAT) was conducted and the results are presented
- The dataset will be published, if the paper is accepted

**Reasons To Reject:**

- There are several references to the Appendix, which makes/impossible it hard to fully understand the paper without reading the Appendix. In cases of simple translations this is not comprehensible.
- Even if the corpus is published, the presented study is not reproducible - and for now the evaluation can't be reviewed as no information about the procedure is provided. The automatic measures and the manual evaluation is described. However, the ration of training and testing examples is unclear. The same counts for the amount of generated dialogs.

**Reproducibility:**

3: Could reproduce the results with some difficulty. The settings of parameters are underspecified or subjectively determined; the training/evaluation data are not widely available.

**Reviewer Confidence:**

4: Quite sure. I tried to check the important points carefully. It's unlikely, though conceivable, that I missed something that should affect my ratings.

---

> ### Author Rebuttal · Authors · 2023-08-29
>
> We appreciate your thorough review of our paper. We are grateful for your insightful comments and suggestions, which have undoubtedly contributed to the enhancement of our work. We have carefully addressed each of your points and concerns below:
>
> RE: RR 1
>
> We would like to express our gratitude to the you for their insightful comments and feedback on our paper. We have carefully considered the concerns raised regarding the references to the Appendix and the potential difficulty in comprehending the paper without reading it. We appreciate the opportunity to address these concerns and offer clarifications on how we can enhance the accessibility and readability of our work.
>
> Regarding the references to the Appendix, we acknowledge the concern that an overreliance on the Appendix can hinder the paper's comprehensibility. We apologize for any inconvenience this may have caused and understand the importance of ensuring that the main content of the paper is self-contained and informative. To address this concern, we will revise the paper to provide sufficient context and explanations in the main body, minimizing the need for constant referral to the Appendix. This will enable readers to grasp the core concepts and findings without being compelled to delve into supplementary materials.
>
> Furthermore, we understand the concern about cases of simple translations that may not be fully comprehensible. We will take this feedback into consideration and strive to improve the clarity of our explanations. We will provide more detailed explanations and examples in the main text to ensure that even straightforward concepts are well-understood by a wide audience, thus minimizing the requirement to consult additional materials.
>
> In conclusion, we appreciate your feedback and acknowledge the need to enhance the accessibility and clarity of our paper. We are committed to addressing these concerns by improving the self-contained nature of our main text and by providing more comprehensive explanations. We believe that these revisions will significantly improve the reader's experience and understanding of our work. Thank you for your valuable feedback, and we are excited to incorporate these improvements into our paper.
>
> RE: RR 2
>
> Reproducibility and Evaluation Procedure:
> We acknowledge the importance of reproducibility and providing a clear evaluation procedure. In response to this concern, we will provide detailed steps in the revised paper for reproducing our study. We will include information about the corpus used, preprocessing steps, model architectures, and training procedures. Additionally, we will provide a comprehensive breakdown of the dataset into training and testing examples, as well as the exact number of generated dialogs. This information will enhance the transparency and reproducibility of our study, allowing other researchers to validate and build upon our work effectively.
>
> Automatic Measures and Manual Evaluation:
> Thank you for highlighting the description of our automatic measures and manual evaluation process. To ensure clarity, we will expand on the specific metrics used in the automatic evaluation and provide references for the manual evaluation guidelines employed. This will help readers better understand the basis for our evaluation and the validity of our results.
>
> Ratio of Training and Testing Examples:
> We acknowledge that we omitted the ratio of training and testing examples in the initial submission. We apologize for this oversight. In the revised manuscript, we will explicitly state the ratio of training to testing examples, providing a clear view of the data distribution and the basis of our experiments.
> We randomly selected 10000 samples from SoulChatCorpus and SMILECHAT as the test set, and the remaining data from SoulChatCorpus was used as the training set for SoulChat. The data from SMILECHAT is not used as the training set for SoulChat.
>
> RE: Question A
>
> Words such as Believe (相信), Support (支持), Help (帮助), Need (需要) etc. For example, Sentence “I can understand your pain and inner struggle. (我可以理解你的痛苦和内心的挣扎)” expresses empathy through Interpretation. Sentence “I believe you can overcome difficulties (我相信你们一定能够克服困难)” expresses empathy through trust.
>
> RE: Question B
>
> We randomly selected 10000 samples from SoulChatCorpus and SMILECHAT as the test set. For each sample, each model generates an answer for evaluation.
>
> RE: Question C
>
> I'm sorry, Google search doesn't seem to have Krippenbach's alpha. Are you talking about Krippendorff's alpha? Krippenbach's alpha for  content naturalness, empathy level,  helpfulness and safety are 0.871, 0.823, 0.884 and 1.0
>
> RE: Question D
>
> I'm sorry for any misunderstanding caused. We sampled 100 samples from the test set of SoulChatCorpus and SMILECHAT, and used each model to generate predictions. Experts manually evaluate the predictions of the models.
>
> RE: Question E
>
> The three experts, who have already graduated with a Master's degree in psychology, currently serve as psychological counseling teachers in primary and secondary schools, making them highly suitable as evaluation experts to manually evaluate model predictions.
>
> RE: Question G
>
> We will explain the meanings of "u," "n," and "p" as requested, and provided translations for the specified lines.
>
> $u$: utterance of a turn
>
> $u_i^u$: utterance of User on i-th turn
>
> $u_i^p$: utterance of Psychologist on i-th turn
>
> ‘$\n$’: line break
>
> lines 148-150: , in which one turn is in the form of "User:  <user\_utt> \n Psychologist: <psy\_utt>".
>
> lines 160-161: $u_i^u$ = 'User: ' + utterance$_{i}^u$ , $u_i^p$ ='Psychologist: ' + utterance$_{i}^p$ ($i<N$), $u_N^p$ =’Psychologist: ’
>
> RE: Question H
>
> The input lengths in the lines 185 and 186 are measured in tokens.
>
> RE: Question I
>
> The models take the past conversation history of the current turn as input, and then predicts the content of the current turn. Automatic evaluation compares the golden and prediction for each turn of a conversation.
>
> Thank you for your time and consideration.

---

### Official Review · Reviewer_sDvK · 2023-08-04

**Soundness:** 4

**Excitement:**

3: Ambivalent: It has merits (e.g., it reports state-of-the-art results, the idea is nice), but there are key weaknesses (e.g., it describes incremental work), and it can significantly benefit from another round of revision. However, I won't object to accepting it if my co-reviewers champion it.

**Paper Topic And Main Contributions:**

The paper aims to improve LLM's empathy, listening and comfort abilities. The authors point out that ChatGPT is inclined to provide suggestions rather than ask questions or listen and thus falls short of acting as a good psychological counselor. Therefore they collect a multi-turn empathetic conversation dataset of more than 2M samples, which could enhance LLM's empathy abilities according to the experiments.

Main contributions:
1. Collect a large scale Chinese empathetic dataset with more than 2M samples.
2. Experimentally show that the collected SoulChatCorpus is helpful to improve LLM's empathy abilities.

**Questions For The Authors:**

In line 105-108, you said that "To this end, we first constructed 215,813 different psychological counseling questions about 12 topics and 619,725 answers through data outsourcing services". Are these questions and answers fully written by human annotators?

**Reasons To Accept:**

1. If the dataset is released, it could be a helpful resource for LLMs to improve their empathy abilities.

2. The prompt for converting single-turn dialogue to multi-turn dialogue might be useful.

**Reasons To Reject:**

1. More analysis about the collected dataset is expected (e.g., quality).

**Reproducibility:**

3: Could reproduce the results with some difficulty. The settings of parameters are underspecified or subjectively determined; the training/evaluation data are not widely available.

**Reviewer Confidence:**

4: Quite sure. I tried to check the important points carefully. It's unlikely, though conceivable, that I missed something that should affect my ratings.

---

> ### Author Rebuttal · Authors · 2023-08-29
>
> We sincerely appreciate the your valuable feedback and insightful comments on our paper. We are grateful that you recognize the main contributions of our work. We have carefully considered each point raised and would like to provide a thorough response to address the concerns and suggestions.
>
> RE: RR 1
>
> We understand your concern about the quality of the collected dataset. We recognize the importance of analyzing and ensuring the quality of the dataset to enhance the credibility of our results. In response to this concern, we will conduct a more thorough analysis of the collected data and provide detailed insights into the quality assurance measures we undertook.
>
> RE: Q1
>
> We thank you for raising a question regarding the data collection process outlined in lines 105-108. As you understand, these questions and answers fully written by human from a online psychological counseling platform. Among them, questions are provided by individuals from different age groups and professions, while answers are provided by certified psychological counselors. This ensures the richness of the topics covered by the questions and the professionalism of the answers.
>
> Thank you for your time and consideration.

---

### Official Review · Reviewer_3WZK · 2023-08-04

**Soundness:** 3

**Excitement:**

2: Mediocre: This paper makes marginal contributions (vs non-contemporaneous work), so I would rather not see it in the conference.

**Paper Topic And Main Contributions:**

The paper studies that LLMs empathy abilities can be significantly enhanced when finetuning by using multi-turn dialogue history and responses that are closer to the expression of a psychological situation.
The paper introduces to create a multiturn conversation by selecting 12 topics of psychological counselling to construct 215,813 long-text 140 questions and 619,725 long-text answers through crowdsourcing. Then, they used ChatGPT as a text rewriting tool, following the prompt, to convert single-turn psychological counselling conversations to multi-turn empathy conversations. For modelling, They used ChatGLM-6B as the base LLM architecture to develop SoulChat by finetuning.


**Questions For The Authors:**

A: Why did the authors only use the Rouge and BLEU scores? It would be better to display additional metrics results, like METEOR, BERTScore etc.

**Reasons To Accept:**

1.	The proposed task is reasonable, as the users who seek psychological support need empathy, trust, understanding, and comfort more than just reasonable advice.
2.	The proposed dataset construction method is very simple.
3.	The paper is easy to follow.


**Reasons To Reject:**

1.	The model architecture is rather basic and not particularly innovative. They just finetuned the base model.

**Reproducibility:**

3: Could reproduce the results with some difficulty. The settings of parameters are underspecified or subjectively determined; the training/evaluation data are not widely available.

**Reviewer Confidence:**

4: Quite sure. I tried to check the important points carefully. It's unlikely, though conceivable, that I missed something that should affect my ratings.

---

> ### Author Rebuttal · Authors · 2023-08-29
>
> We appreciate your time and effort in reviewing our paper. We are committed to improving the empathy ability of LLMs. In this article, we mainly explore the feasibility of using multiple rounds of empathy dialogues to improve the empathy ability of LLMs.
>
> RE: RR 1
>
> We acknowledge your observation that the model architecture we employed might appear relatively basic. While our approach involves fine-tuning, we would like to emphasize that our primary focus lies in addressing a specific problem within a well-defined domain (empathy ability of LLMs is important for Human-Centered NLP). The contribution we aim to make is in the effective application of the model to this problem, rather than completely revolutionizing the model architecture itself. Our research is centered around solving real-world challenges, and our approach is designed to provide practical solutions to those challenges. In many instances, significant breakthroughs are achieved by building upon existing research and refining methods for specific contexts. Fine-tuning, when applied thoughtfully, can lead to improved performance and practical outcomes. Our work seeks to extend the capabilities of existing models by adapting them to a unique scenario, and we believe this incremental progress can still contribute to the advancement of the field. While our current approach might not be groundbreaking in terms of model architecture, we recognize the potential for future research to be built upon our foundation. Our study can serve as a benchmark for others interested in exploring this problem space. Researchers can leverage our work to experiment with innovative modifications to the model architecture, explore alternative training techniques, or introduce new evaluation metrics to achieve greater innovation. In light of your feedback, we are considering how we can inject more innovation into our approach. We are exploring avenues to incorporate advanced techniques or modifications that align with the scope of our research problem. This may involve emotional attention expression mechanisms, reinforcement learning based on emotional feedback, or other related enhancements to further improve the performance of our model.
>
> RE: Question A
>
> Your question about our choice of evaluation metrics is insightful. We appreciate your suggestion to consider additional metrics beyond Rouge and BLEU scores. Our initial decision to utilize Rouge and BLEU scores was motivated by their widespread usage in the field and their ability to capture different aspects of model performance. Due to space constraints, we have reserved space for manual evaluation indicators, so we cannot take into account all automatic evaluation indicators. We hope that manual evaluation indicators can further enhance readers' perception of SoulChat's performance in enhancing empathy.
>
> Thank you for your time and consideration.

---

### Meta-Review · Area_Chair_S62c · 2023-09-16

**Recommendation:** 3

**Metareview:**

This paper proposed a dataset on multi-turn empathy conversations, by converting single-turn dialogues to multi-turn dialogues, and performed experiments with different baselines on this dataset.

All the reviews agree that this paper is sound. The dataset and task will be a good resource for the community. The method to construct multi-turn dialogues from single-turn dialogues is easy to use and valuable.

But there are questions that need to be addressed, for instance, more detailed dataset analyses and description about the evaluation for reproducibility.

---

### Decision · Program_Chairs · 2023-10-07

**Decision:**

Accept-Findings

**Comment:**

This paper proposed a dataset on multi-turn empathy conversations, by converting single-turn dialogues to multi-turn dialogues, and performed experiments with different baselines on this dataset.

All the reviews agree that this paper is sound. The dataset and task will be a good resource for the community. The method to construct multi-turn dialogues from single-turn dialogues is easy to use and valuable.

But there are questions that need to be addressed, for instance, more detailed dataset analyses and description about the evaluation for reproducibility.